# Self-supervised visual place recognition for colonoscopy sequences

**Javier Morlana**                                                       JMORLANA@UNIZAR.ES

**Pablo Azagra**                                                         PAZAGRA@UNIZAR.ES

**Javier Civera**                                                        JCIVERA@UNIZAR.ES

**J. M. M. Montiel**                                                     JOSEMARI@UNIZAR.ES

*Universidad de Zaragoza*

## Abstract

We present the first place recognition system trained specifically for colonoscopy sequences. We use the convolutional neural network for image retrieval proposed by Radenović et al. and we fine-tune it using image pairs from real human colonoscopies. The colonoscopy frames are clustered automatically by a Structure-from-Motion (SfM) algorithm, which has proven to cope with scene deformation and illumination changes. The experiments show that the system is able to generalize by testing in a different human colonoscopy, retrieving frames observing the same place despite of the different viewpoint and illumination changes. The proposed place recognition would be a key component of Simultaneous Localization and Mapping (SLAM) systems operating in colonoscopy to assist doctors during the explorations or to support robotization.

**Keywords:** Deep Learning, Place Recognition, CNNs, Colonoscopy

## 1. Introduction

Despite the advances on technology and computer vision, colonoscopies are still performed in a very manual manner, mainly the doctors inspecting the images of the endoscope camera. SLAM systems track the camera pose while creating a map of the environment, which could assist doctors in clinical decisions and treatment delivery. Colonoscopies, however, present huge challenges for SLAM, like low-texture areas or occlusions caused by fluids or tools during the exploration, producing tracking failures and incomplete or disconnected maps.

Having a database of previously observed frames, Visual Place Recognition (VPR) refers to the capability to retrieve frames depicting the same scene region than the query. In the context of SLAM, VPR is a key component in camera relocalization when the tracking is lost, loop closing and map reuse. VPR could also help doctors in recognizing the place where an injury was located in a previous exploration.

Traditional VPR methods in SLAM are based on Bag-of-Words (BoW), being (Mahmoud et al., 2016) the first to apply it for endoscope sequences. Recently, Convolutional Neural Networks (CNNs) outperformed traditional approaches in the task of VPR, with (Radenović et al., 2018) being one of the most relevant works. They formulate explicitly VPR as a particular case of image retrieval and describe images by means of a global descriptor obtained after a forward pass through the CNN.

This work is a first step to bring these VPR methods to SLAM for colonoscopies. Specifically, we fine-tune the architecture proposed in (Radenović et al., 2018) for the specific domain of colonoscopies, giving place to the first VPR method in the colon.

## 2. Self-supervised training of the visual place recognition network

Our goal is, given a query image from a colonoscopy, to retrieve other frames from that colonoscopy that observe the same place. We select 5 different real human colonoscopies, using 3 of them for training, one for validation and the last one for testing. As the sequences are unlabelled, we need to identify the covisible frames that would give positive examples for training using the contrastive loss. We chose the state-of-the-art SfM pipeline COLMAP (Schonberger and Frahm, 2016) that creates up to 50 independent 3D clusters for every sequence by the use of SIFT features and 3D geometry. We perform exhaustive matching between all the frames of the 40 fps sequence. This expensive computation allows to overcome the challenges like fluids, deformations and illumination changes.

We chose the network model proposed in (Radenović et al., 2018), with the ResNet101 architecture (only convolutional layers, fully-connected layers are discarded), followed by Generalized-Mean (GeM) pooling and L2-normalization. The result after a forward pass of an image is a global 2048-dimensional image embedding or descriptor, which is used to compute the image similarity by means of the inner product.

The SfM clustering yields about 20k images for training and 8.5k for validation. Positive pairs are frames from the same 3D cluster with at least one covisible 3D point. We randomly select 4k query-positive (q-p) pairs to ensure variability in the data. For every q-p pair, 5 hard-negatives are selected from the same sequence online during training, choosing as hard-negative frames those with a similar descriptor. Each negative example comes from a different 3D cluster, excluding the query cluster. Training is performed for 100 epochs, keeping the best model that achieves minimum loss on the validation set. Images are resized to a maximum size of $362 \times 362$ while maintaining their aspect ratio.

## 3. Results and conclusions

We compare our colonoscopy fine-tuned network with the best model reported in (Radenović et al., 2018), which was trained with images of popular landmarks and cities. The 6k images test set is composed of the frames that belongs to a 3D cluster after SfM processing. The database includes all the test images and the covisible images ground truth is generated by COLMAP. Every database image is used as a query and its descriptor is compared against the rest of the database except itself. Frame rate extraction is 10 fps so consecutive images are separated by 0.1 seconds.

For evaluating the performance, we compute Recall@$N$. This is the recall obtained when varying the number of top $N$ candidates retrieved, understanding recall as the percentage of queries that retrieve at least one correct prediction within the first $N$ candidates. To evaluate the system performance in relating images far from the query, which present challenging viewpoint and illumination changes, we add a minimum inter-frame distance threshold $D$ to retrieve only images separated from the query by at least $D$ frames, i.e. if $D = 5$, only images separated from the query by 5 frames (0.5 seconds) or more are retrieved.

Table 1 shows the Recall@$N$ values for different values of the threshold $D$. It can be seen that our method outperforms the original network in all cases, while the relative difference between methods increases when the $D$ parameter increases, showing that our approach is able to retrieve more widely separated frames. It is remarkable that the first retrieval (R@1) when $D = 15$ is correct in 85.3% of the cases.

Table 1: Recall@$N$ [%] for different values of $D$

|  | D = 5 | | | D = 10 | | | D = 15 | | |
|---|---|---|---|---|---|---|---|---|---|
|  | R@1 | R@5 | R@10 | R@1 | R@5 | R@10 | R@1 | R@5 | R@10 |
| Radenović | 92.2 | 96.0 | 97.3 | 83.7 | 90.3 | 92.7 | 77.3 | 85.6 | 88.7 |
| Our method | **96.1** | **98.2** | **98.7** | **90.5** | **94.9** | **96.5** | **85.3** | **91.3** | **93.4** |

Qualitative results are shown in Fig. 1, where two queries are displayed along with correct and wrong retrievals. For the first case, the network retrieve correct frames presenting different viewpoints and illumination changes. It is noteworthy that the query and some retrievals can be widely separated in time, e.g. $D = 3101$ or $D = 2944$, being able to relate frames between the insertion and the withdrawal. In the second case, the network retrieves frames with different viewpoint but gets confused with images with similar structure.

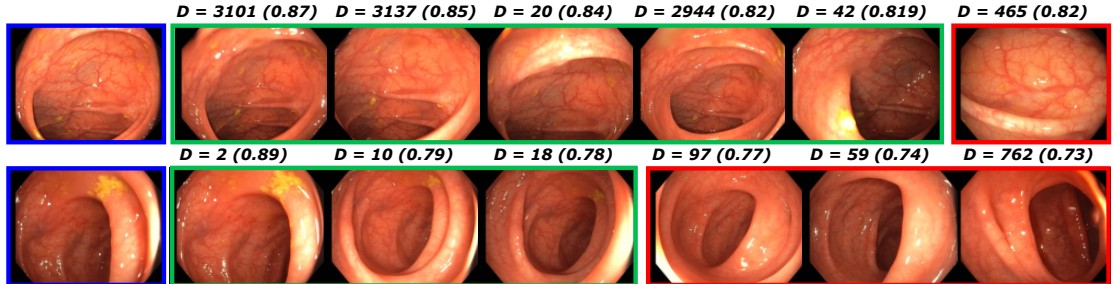

Figure 1: Network retrievals. The query is framed in blue, correct retrievals (same place) in green and wrong retrievals (different place) in red. $D$ is displayed for every query along with the image similarity score in parenthesis.

In conclusion, this work presents the first fully automated VPR system fine-tuned for the colon domain, which is a key step towards building a full SLAM system for colonoscopy. A single-query forward pass takes about 15 ms which is suitable for real-time SLAM. Our experiments validate the approach, the next steps will be focused on improving the performance by increasing the amount of training data, designing novel architectures that exploit the particular domain of colonoscopies or applying spatial and temporal verification.

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
