# OpenReview forum: "Self-supervised Visual Place Recognition for Colonoscopy Sequences"
_MIDL.io/2021/Conference/Short — MIDL 2021 Poster_

### Official Review · Reviewer_5WRt · 2021-04-21

**Confidence:** 5
**Final Rating:** 1

**Summary:**

This work reported an application of visual-place-recognition (VPR) technique to colonoscopic images. The authors fine-tuned CNN-based VPR method (Radenovic et al., 2018) and reported the results. From its low readability, detail contribution is unclear. In my understanding, the authors clustered frames of colonoscopic videos into 50 independent clusters by SfM piepline COLMAP and defined them as 50 locations.  The authors then classified queries by the fine-tuned CNN into one of 50 locations.

**Strengths:**

Challenging the application of VPR technique to colonoscopic images. In 3D reconstruction of a colon from 2D colonoscpic images,  accurate image matching between colonoscpic scene is quite difficult.

**Weaknesses:**

1. Low readability of manuscript gives unclear methodology and its evaluation procedures. It is hard to understand their contributions.

2. Evaluation setting is not convincing.

The authors clustered frames of colonoscopy videos by SfM pipeline with SIFT features. However,  transformation-invariant local feature s are not valid for colonoscope images. Therefore, the accurate clustering looks impossible. My question is how to defined the 50 locations? Do they have anatomical meaning? And, did authors confirm how correctly clusteree?

**Deanonymize Review:**

no

**Detailed Comments:**

Even though I am familiar with 3D reconstruction with multiple-view geometry and colonosocpic image processing including classical and recent deep learning approaches, this manuscript is hard to understand. I recommend re-organising of manuscript and making clear logical flow with grammatically corrected English.



**Justification Of The Rating:**

As I commented in 'Weaknesses' and 'Detailed Comments', the low readability of submitted manuscript makes the authors' contributions unclear. For example, the authors described 'We chose the state-of-the-art SfM pipeline COLMAP (Schonberger and Frahm, 2016) that creates up to 50 independent 3D clusters for every sequence by the use of SIFT features and 3D geometry.' in section 2.  However, a reader can not understand the reason why the authors adopted this method for what?  I concluded that this manuscript is premature for the submission.

**Paper Type:**

methodological development

**Special Issue:**

no

---

### Official Review · Reviewer_EEpS · 2021-04-30

**Confidence:** 5
**Final Rating:** 3

**Summary:**

This paper addresses the problem of -- how to improve visual place recognition in colonoscopy data. Authors employed tools drawn from image retrieval and SfM.  The main contribution of this paper is the application side. Authors addressed a challenging problem building upon existing techniques
for the particular case when using colonoscopy sequences.


**Strengths:**

-- The main highlight of this paper is the application side (no technicality is presenting in the current paper). Authors deal with a challenging scenario where several complexities need to be considered including occlusions and specular reflections.






**Weaknesses:**

-- The technical contribution is limited -- therefore, the paper is appreciated as a case study with limited novelties. The authors strongly build upon existing techniques. There is no clear discussion on the potential technique contribution.

-- Whilst authors indeed offer initial experiments showing the potentials for the application, the discussion displayed in the paper makes it difficult to read a conclusion regarding the application. This makes the paper to be perceived with limited advantages.


**Deanonymize Review:**

no

**Detailed Comments:**

-- Whilst the application is interesting, there is no technical novelty in the paper. Authors fail to transmit  the technical difficulties of the application.  The techniques of that [Schonberger] and [Radenovic] has been used extensively for other types of data showing potential results.   This makes the paper to be perceived as a only a case study with limited advantages. For example, authors strongly claim that this is specifically for colonoscopy sequence -- however, general endoscopic data can be used in the framework -- authors should clarify their claim that it is only for colonoscopy sequences.-- what is special when handling this type of data in comparison with any general endoscopic procedure?

-- Can the authors rapport the hyperparameters used? Beyond only mentioning the number of epochs.

--This is a short paper and not a large number of experiments are expected. The authors reported initial results - however, the authors fails to transmit the main findings in the paper.  A better discussion on the findings of all results will strengthen the paper.



**Justification Of The Rating:**

This work is an application paper with no technical novelty. The application side is welcome.  This is a short paper that includes a good balance between initial experiments and providing the big picture of the problem. Because of these reasons, I suggest a weak accept.

**Paper Type:**

validation/application paper

**Special Issue:**

no

---

### Meta-Review · Area_Chair_izCd · 2021-05-07

**Recommendation:** Accept (Poster)
**Confidence:** 5

**Metareview:**

While the AC agrees that there are some shortcomings in the details of the presentation, the AC does find the presentation to be acceptable and agrees with reviewer EEpS that this is an interesting application; therefore the paper can be accepted as an application paper.

---

### Decision · Program_Chairs · 2021-05-11

Accept (Poster)